# Microisolation of Spatially Characterized Single Populations of Neurons for RNA Sequencing from Mouse and Postmortem Human Brain Tissues

**DOI:** 10.3390/jcm12093304

**Published:** 2023-05-05

**Authors:** Melissa J. Alldred, Stephen D. Ginsberg

**Affiliations:** 1Center for Dementia Research, Nathan Kline Institute, Orangeburg, NY 10962, USA; 2Department of Psychiatry, New York University Grossman School of Medicine, New York, NY 10016, USA; 3Department of Neuroscience & Physiology, New York University Grossman School of Medicine, New York, NY 10016, USA; 4NYU Neuroscience Institute, New York University Grossman School of Medicine, New York, NY 10016, USA

**Keywords:** RNA-seq, laser capture microdissection, selective vulnerability, bioinformatics, single population, spatial profiling

## Abstract

Single-cell and single-population RNA sequencing (RNA-seq) is a rapidly evolving new field of intense investigation. Recent studies indicate unique transcriptomic profiles are derived based on the spatial localization of neurons within circuits and regions. Individual neuronal subtypes can have vastly different transcriptomic fingerprints, well beyond the basic excitatory neuron and inhibitory neuron designations. To study single-population gene expression profiles of spatially characterized neurons, we have developed a methodology combining laser capture microdissection (LCM), RNA purification of single populations of neurons, and subsequent library preparation for downstream applications, including RNA-seq. LCM provides the benefit of isolating single neurons characterized by morphology or via transmitter-identified and/or receptor immunoreactivity and enables spatial localization within the sample. We utilize unfixed human postmortem and mouse brain tissue that is frozen to preserve RNA quality in order to isolate the desired neurons of interest. Microisolated neurons are then pooled for RNA purification utilizing as few as 250 individual neurons from a tissue section, precluding extraneous nonspecific tissue contaminants. Library preparation is performed from picogram RNA quantities extracted from LCM-captured neurons. Single-population RNA-seq analysis demonstrates that microisolated neurons from both postmortem human and mouse brain tissues are viable for transcriptomic profiling, including differential gene expression assessment and bioinformatic pathway inquiry.

## 1. Introduction

For the last several decades, even prior to the development and implementation of RNA sequencing (RNA-seq) technologies, researchers have utilized various experimental protocols to query RNA expression levels of specific genes, using methods such as qPCR, microarray analysis, and chip-based technologies, which all strive to resolve the burning question, “what genes are differentially expressed?” [1,2,3,4,5,6,7,8]. While many of these technologies are still in use today and show a high overlap of gene expression results with RNA-seq [9], they all have drawbacks, including the number of genes that can be reasonably queried, cost, amount of starting material needed, and sequence specificity, especially when quantitating highly homologous genes [10,11].

In the last decade, a concerted effort has been put forth to advance RNA-seq technologies, which have several key benefits over previous transcriptomic identification and quantification technologies, including the ability to detect previously unidentified noncoding RNA (ncRNA) species, identification of sequence variations, including single nucleotide polymorphisms (SNPs) and highly homologous sequences, as well as having low to no background [12]. There are several methodologies for examining gene expression profiling within the brain, including bulk tissue involving several brain regions, individual regional dissections, and profiling involving single-cell technologies. These RNA-seq approaches have increased the ability to examine gene expression profiles in single populations and in single cells from brain tissues. However, there are several caveats to understanding differentially expressed genes (DEGs) in these cells. Single-cell RNA-seq in the brain can be a misnomer, as many of these studies in adult brains are based on single nuclei, not intact cells [13,14,15], or they may use mechanical or enzymatic homogenization approaches to isolate individual cells [16,17,18]. Mechanical and enzymatic dissociation have additional technical difficulties, as enzymatic dissociation of fresh tissue can affect the gene expression profile, especially in regard to inflammatory markers, and mechanical dissociation can result in very few neurons isolated, as their long axonal and dendritic processes cause shearing and tearing [17,19,20]. Further, these samples are typically isolated from admixed brain regions, or if they are from individual brain regions, are from admixed neuronal and non-neuronal cell types that need to be identified post hoc by deconvolution of the resulting RNA-seq gene expression profile [15,16,17,21]. In brain regions that are difficult to isolate by fiducial landmarks due to location and/or size of the region, the number of neurons quantified in these regions is limited [15,22], and alternate methodologies such as Patch-Seq are often employed, which are highly labor-intensive and prone to experimenter error [23]. In disease models where little is known about the gene expression profiles of vulnerable individual cell types, changes in gene expression due to disease may overlap with changes live cells undergo during isolation, all of which can cause variability in gene expression profiles and loss of critical data for mechanistic pathway identification. Studies have been undertaken to examine this quandary, but limitations exist in determining cell identity past categories, such as excitatory or inhibitory neurons, microglia, and astrocyte designation [24,25].

Recently, high-precision technologies have improved enough to examine spatial transcriptomic profiles, including laser capture microdissection combined with RNA-sequencing (LCM-Seq) [26], spatial transcriptomics [27], and *in situ* sequencing [28]. Technical difficulties still arise from these technologies, including the number of genes curated, spatial resolution, and fixation requirements, all of which limit resolution. Taking into account the potential as well as the limitations of current single-population technologies in the context of transcriptomic profiling, we developed a methodology combining LCM and RNA-seq from fresh frozen tissue sections, which obviates the degradation of RNA from fixation protocols and can be performed on individually identified neuronal subtypes from spatially mapped brain tissue from animal models as well as postmortem human brain tissues.

## 2. Materials and Methods

### 2.1. Tissue Accrual

Animal protocols were approved by the Nathan Kline Institute/New York University Grossman School of Medicine IACUC in accordance with NIH guidelines. In order to preserve the RNA quality and quantity for single neuron dissections, neurons from mouse and postmortem human tissue were isolated from unfixed frozen tissue. Mice were anesthetized with ketamine (13 mg/kg, Henry Schein, Melville, NY, USA) and xylazine (80 mg/kg, Henry Schein, Melville, NY, USA). Once mice were unresponsive to toe pinch, tails were taken to confirm genotype, and chest cavity was opened, wherein mice were checked for tumors or other malformations. Mice were perfused transcardially (Cole-Palmer Masterflex L/S, Vernon Hills, IL, USA) with a 22-gauge catheter using 0.15 M phosphate buffer (pH 7.2) at 7.5 mL/minute (min) for ~2–3 min until the liver was completely cleared of blood. Following completion of perfusion, brains were removed from the skull and immediately flash frozen with powdered dry ice or liquid nitrogen. Once brains were frozen (Figure 1A,B), they were labeled and placed in −80 °C for storage until cut for tissue collection.

### 2.2. Cryocutting of Spatially Identified Samples

Utilizing either postmortem human (Figure 1E) or mouse brain tissue (Figure 1A,B), brains were stored on dry ice to equilibrate while mounting. Slides for LCM were prepared as follows: racks (capable of holding 25 slides) of polyethylene naphthalate (PEN) or polyethylene terephthalate (PET) membrane slides (Leica Microsystems, Wetzlar, Germany) were processed for RNase deactivation by dipping them for 10 min in RNase Away (7005-1, ThermoFisher, Waltham, MA, USA), then rinsed twice in 18.2 megaohm water (Barnstead Nanopure, ThermoFisher, Waltham, MA, USA) before air drying completely. Once dry, individual slides were put back into their original boxes precleaned with RNase Away. In the cryostat (−20 °C), a thin layer of Tissue Tek OCT Compound (ThermoFisher, Waltham, MA, USA) was spread on the specimen disk. Once the OCT became tacky, the brain was placed on the OCT and oriented for the best cutting surface using precooled forceps. The specimen disk was placed on the quick freeze shelf in cryostat and allowed to harden (Figure 1C,F). Additional OCT was layered along the base of the tissue for additional support. Sections were cut at 20 µm using the antiroll plate and adhered to a 4 µm PEN or 0.9 µm PET membrane slide while in the cryostat (Figure 1D,G). Once cut tissue was on the slide, tissue was stored in prechilled (on dry ice) True North (safe for cryostorage; HS15988H, Heathrow Scientific, Vernon Hills, IL, USA) slide boxes with a desiccant pouch (Drierite or silica gel desiccant; ThermoFisher, Waltham, MA, USA) inside each box. Slide boxes were then removed to prechilled (−80 °C) weathertight totes (Iris #110380; Staples, Framingham, MA, USA) and stored at −80 °C until use.

### 2.3. Sample Preparation for Laser Capture Microdissection

True North boxes capable of holding 25 slides (HS120316, Heathrow Scientific, Vernon Hills, IL, USA) with a small, perforated bag of desiccant were equilibrated at −80 °C, then put on dry ice. To select slides for LCM, a single slide box of stored cryocut samples was removed from weathertight totes and put on dry ice. Slides were chosen quickly and moved to 25-place True North boxes. Once completed, both slide boxes were returned to −80 °C in weathertight tote until ready to perform staining or LCM. To perform staining or LCM, the True North boxes with the selected specimens were equilibrated stepwise to room temperature (RT) as follows: closed slide boxes were quickly moved from −80 °C to −20 °C and incubated for 5 min. After 5 min equilibration at −20 °C, the slide box was quickly moved to 4 °C for 10 min, then moved to RT for 5 min without opening True North boxes with desiccant inside. Once equilibrated to RT for 5 min, boxes were opened, and slides were stained.

### 2.4. Antibody Staining

A rapid protocol based on the Vector Labs ImmPress HRP reagent kit (anti-goat MP-7405, Vector Labs, Newark, CA, USA) was used, with the ImmPACT NovaRed chromogen employed as the peroxidase substrate (SK-4805, Vector Labs, Newark, CA, USA). To preserve the RNA from the tissue, phosphate-buffered saline (PBS, pH 7.2), blocking buffer (from the ImmPress Kit, Vector Labs, Newark, CA, USA), diluent buffer, and secondary antibody solutions (from the ImmPress kit, Vector Labs, Newark, CA, USA) had 1 µL (20 U) per ml of RNase Inhibitor (AM2696, ThermoFisher, Waltham, MA, USA) added to solution immediately prior to use. Briefly, stepwise equilibration was performed, and slides were rinsed with ice-cold PBS. Blocking buffer was incubated for 5 min at RT (from the ImmPress kit, NHS blocking buffer). Blocking buffer was decanted, and primary antibody solution was added. To identify basal forebrain cholinergic neurons [29,30], we utilized an anti-choline acetyltransferase (ChAT) antibody (1:50, AB-144P, Sigma-Aldrich, St. Louis, MO, USA) in ice-cold diluent buffer (1× PBS with 2% normal goat serum (Vector Labs, Newark, CA, USA), 0.01% Triton X-100 (Sigma-Aldrich, St. Louis, MO, USA) and 1 µL RNase Inhibitor per mL), which was incubated for 25 min at RT in humidified chamber to reduce evaporation. Slides were rinsed with PBS twice with ~500 µL per slide. Cold secondary antibody solution was added dropwise onto the slide with a 1 mL pipette (premade, aliquoted into 1.5 mL tube, and RNase Inhibitor added) and incubated at RT 20 min in humidified chamber. During secondary antibody incubation, NovaRed peroxidase substrate was made per manufacturer’s specifications and kept at 4 °C until ready to use. Once secondary incubation was complete, slides were rinsed with PBS twice with ~500 µL per slide. NovaRed solution was added to slides and incubated for 5 min at RT. Slides were rinsed with PBS twice (~500 µL per slide). Slides were then air dried at RT for ~5 min or until the slide was completely dry. Slides were used directly for LCM or stored under desiccant at −80 °C until ready for LCM (Figure 2C,D). Detailed protocols for staining are available in Appendix A.

### 2.5. Nissl Staining

Briefly, stepwise equilibration was performed, and slides were rinsed with ice-cold PBS with optional 1 µL RNase Inhibitor added per ml of solution. Nissl stain was added (49.44 mM sodium acetate, 3.6 mM glacial acetic acid, 0.1% thionin) for ~30 s^−1^ min at RT [31]. Slides were rinsed twice with ice-cold PBS and air dried for ~5 min or until slide was completely dry. LCM was performed immediately (Figure 2A,B,F).

### 2.6. Laser Capture Microdissection

Samples were moved immediately following staining or equilibration to the LCM apparatus (LMD7000, Leica Microsystems, Wetzlar, Germany). LCM was limited to 1 h at RT to reduce RNA degradation. The ‘specimen overview’ feature (Leica Microsystems, Wetzlar, Germany) was used to generate a quick overview image using the 1.25× or 5× objective to orient tissue sections for cutting of single cells. Neurons were collected in 50 µL of ice-cold Qiazol (Qiagen, Germantown, MD, USA), utilizing the 40× (N.A. 1.4) or 20× (N.A. 0.85) objective using the ‘draw and cut’ feature (Leica Microsystems, Wetzlar, Germany). Stained cell bodies were identified with a Wacom digitizer pen (Leica Microsystems, Wetzlar, Germany), outlined with the ‘point-to-point’ feature or with the ‘automated detection module’ (Leica Microsystems, Wetzlar, Germany) using stained cells to identify cell bodies (outline was set at 15 pixels outside of the identified cell). Cells were outlined slightly outside of the cell body edge to allow for laser cutting line (~3–5 µm on 40× objective, ~6 µm on 20× objective). Offset, power, aperture, speed, specimen balance, head current, and pulse frequency were adjusted as necessary to allow for thin cut line without burning adjacent tissue. If necessary, the ‘laser screw’ feature (Leica Microsystems, Wetzlar, Germany) was utilized with 1–2 steps at 3–5 µm to fully cut cells. Samples were identified by cell type, brain region, and sample number and stored at 4 °C. Once all samples from the slide were microisolated, accuracy of cutting was performed, either by counting collected cells that were fully cut or by using the ‘shape list summary’ feature (Leica Microsystems, Wetzlar, Germany). Samples were briefly spun down and moved to −80 °C until ready for RNA isolation.

### 2.7. RNA Purification

For LCM samples of 500–2000 neurons, the miRNeasy micro kit was employed (Qiagen, Germantown, MD, USA). The manufacturer’s specifications were followed except for the following alterations. The optional DNase digestion step was performed with the following amendments: the RWT initial wash (B1) was performed with 700 µL per sample, not the 350 µL recommended; the RWT wash step B4 was performed with 600 µL RWT buffer; and flow through was collected at 750 µL to rewash the column (step B5). The DNase digestion was performed twice sequentially, meaning that after step B5, steps B2–B5 were repeated before continuing to original protocol. After the DNase digestion, the RPE buffer wash step was performed (step 11), not skipped as suggested by the kit specifications (Qiagen, Germantown, MD, USA). Finally, for the final elution, after samples had 14 µL of RNase-free water added to column, a 1 min incubation was performed before the centrifugation step to help fully elute all RNA available on the column. Quality control was performed by bioanalysis using the 2100 Bioanalyzer system (Agilent, Santa Clara, CA, USA) in association with the RNA 6000 Pico kit (Agilent, Santa Clara, CA, USA) or the Tapestation (Agilent, Santa Clara, CA, USA) with the high-sensitivity RNA screen tape (Agilent, Santa Clara, CA, USA). This allowed for the quantification of the harvested RNA in pg/µL. For the Pico kit, any sample below ~100 pg/µL, and for the Tapestation, any sample below ~500 pg/µL, will not generate an RNA integrity number (RIN), in which case the DV_200_ was used to estimate RNA quality [32]. DV_200_ measures the percentage of RNA quantified that is larger than 200 base pairs.

### 2.8. RNA-Seq Library Preparation

The SMARTer Stranded Total RNA-Seq kit-Pico input mammalian kit (Takara Bio, San Jose, CA, USA) was used according to the manufacturer’s specifications with the maximum amount of RNA utilized (or ~1 ng of RNA for larger neuron collections). Utilizing the fragmentation step was dependent on the quality of the RNA gathered. RNA derived from antibody-stained tissue was only performed with option 2 (without fragmentation). Additionally, for the newest version tested (v3), dependent on the amount and quality of RNA, PCR 1 (section B) and PCR2 (section E) were modified in terms of cycle number. For highly degraded RNA, PCR1 was 10 cycles, and PCR2 was 16 cycles, whereas for cells collected from Nissl-stained tissue, PCR1 was 10 cycles, and PCR2 was 13–14 cycles, depending on the amount of starting RNA (<1 ng =14 cycles, ~1 ng or above = 13 cycles). The Takara v3 kit (Takara Bio, San Jose, CA, USA) includes unique molecular identifiers for dual indexes, which can result in higher visible primer dimer concentrations (see Figure 3D,E).

## 3. Results

Brain preparation is critical for downstream processing of the tissue, with perfusion using phosphate buffer or PBS utilized herein. Following perfusions, mouse brains were quickly dissected and frozen to preserve RNA quality. There are several quick freeze methods that can be utilized for the preservation of brain specimens. We tested three different protocols for rapid isolation and freezing of brain tissue to preserve RNA quality. Liquid nitrogen was utilized in a glass-walled crucible to perform rapid freezing, which cooled brain dissections within seconds of isolation. However, liquid nitrogen also caused cracking of brain tissue (Figure 1A), likely due to the rapid temperature change. We modified our brain collection to examine two alternate methodologies for rapid freezing, which would still freeze tissues in less than a minute after isolation, but without causing distortion of the brain tissue. This was performed by placing brains on dry ice with a freezing platform covered in aluminum foil and overlaying the brain tissues with powdered dry ice or cooling a container of ethanol in a dry-ice-filled cooler and adding 100% ethanol and dry ice as a slurry directly prior to carefully placing the brain into the slurry until frozen (Figure 1B).

Isolation of frozen postmortem human tissue is often conducted by a biobank or repository. Unless a researcher has access to fresh-cut human postmortem brain tissue and an approved protocol for their brain collection, postmortem human specimens are typically delivered not perfused but frozen and blocked by the neuropathologist for the regions of interest (Figure 1E). Once brain tissue is collected/blocked, sectioning is performed by cryostat cutting (Figure 1C,F). To isolate fully intact individual neurons, section thickness is based on cell type, brain region, and specimen species. For excitatory neuron isolation in both mouse (Figure 1D) and postmortem human (Figure 1G) brain tissues, 20 µm thick sections were of sufficient thickness to isolate individual, fully intact neurons without having a z-plane thick enough for multiple cells. Sample preservation at this stage is best performed by limiting temperature changes and reducing condensation on tissue samples.

To preserve tissue quality and prevent crystal formation in and on cells, a step-down equilibration step is utilized, with desiccant added to prevent moisture formation on brain sections. When performed, we found little to no condensation or crystal formation on sections. For brain regions and cell types that can be easily identified by region and morphology, Nissl staining was utilized (Figure 2A,B,F). Alternatively, utilizing the rapid antibody staining methodology allows for the identification of neuronal subtypes and can be performed on frozen tissue sections (Figure 2C,D). LCM can also be performed on fluorescently labeled tissue sections (Figure 2E). To microisolate single cells without ablating tissue by the laser source, careful adjustment of the power, aperture, and focus needs to be performed for each slide and, in some cases, each tissue section and/or discrete brain region on the slide. Figure 2A,C–E for single-cell isolation were microisolated utilizing a 40× objective, as this allowed for clear isolation. The larger-sized human pyramidal neurons (Figure 2B) and CA1 pyramidal neurons collected by multiple cells per cutting maneuver (Figure 2F) were isolated utilizing a 20× objective. Higher magnification allows for a very thin laser-cut line (~2–4 µm) and can microisolate small neurons precisely. As the LCM must be performed at room temperature, to keep RNase activity to a minimum, a 1-h time limit is set from the start of sampling to when the collected neurons in their collection device are cooled to 4 °C. If samples are dropped into a lysis buffer that prohibits RNase activity, this time can be extended to represent 1 h from the start of sampling until the last neuron is dropped into the collection device.

We utilized the miRNeasy kit (Qiagen, Germantown, MD, USA) for RNA isolation, which performs well for total brain and neuronal LCM samples. Previous work in bulk tissue dissection utilizing RNA purification kits without a phenol–chloroform extraction clogged the columns, including the RNeasy kit (Qiagen, Germantown, MD, USA) and the Total RNA mini kit (IBI Scientific, King of Prussia, PA, USA). For LCM samples, an estimate is ~1–10 pg of RNA per neuron. Antibody staining will skew the amount of RNA toward the lower end (Figure 3C; left panel), while the rapid Nissl protocol will skew it higher (Figure 3A,B; mouse, Figure 3D,E; human; left panels). Neurons collected from mouse brains were observed to have less total RNA (Figure 3A–C; left panels) compared to postmortem human brains (Figure 3D,E; left panels) when examining the same number of neurons, although a statistical comparison was not performed.

Takara Bio offers two cDNA library preparation kits that rely on random hexamers as initiation primers, which are also considered low input (~250 pg minimum RNA). These include the SMARTer Stranded Total RNA-Seq kit-Pico input mammalian and the SMART-SEQ Stranded Kit (Takara Bio, San Jose, CA, USA). We utilized both kits per the manufacturer’s specifications and found them to work with mouse (SMARTer) and human (SMARTer and SMART) LCM-collected samples, with at least 500 mouse neurons (Figure 3A–C; right panels) or 250 human neurons (Figure 3D,E; right panels) collected per sample and the maximum volume of RNA input allowed per specifications. When limiting the neuron number to 500 neurons, antibody staining from the mouse tissue reduced the success rate of the cDNA library preparation to ~70% of the samples passing QC for the library (Figure 3C; right panel, SMARTer stranded kit v2, Takara Bio, San Jose, CA, USA).

## 4. Discussion

We illustrated the process of performing LCM on frozen tissue sections with antibody or colorimetric staining to isolate single populations of neurons for subsequent single-population RNA-seq analysis. We identified and tested methodologies that consistently work for multiple cell types from several species, including tissue degraded from antibody-stained tissue, to identify specific neuronal subtypes *in vivo*. During specimen isolation, the perfusion protocol was employed to clear blood contaminants from peripheral tissues, helping to reduce variability with downstream genomics applications. We found freezing specimens with liquid nitrogen led to cracking, and breakage of the tissue occurred due to rapid cooling. This unwanted externality may be reduced with large volume containers, where the brain specimen would not contact the container walls during the cooling process. However, we undertook alternate preservation methods to avoid any tissue cracking concerns. Powdered dry ice and ethanol/dry-ice baths provided rapid freezing without damaging the specimens. These two methodologies were indistinguishable for timing, brain preservation, and RNA quality and quantity generated.

LCM enables microisolation of individual cell types [30,33,34] and circumscribed regions (e.g., ribbons of layer III and/or layer V of cerebral cortex) [35,36,37]. However, extra care must be taken for optimal tissue preparation (Table 1). While the present experiments utilized 20 µm thick cryostat sections, this was based on the average cell thickness of excitatory neurons within mouse and human brain tissue [38,39]. For larger cell types, such as cerebellar Purkinje neurons, especially in human tissues, thicker (30–40 µm or more) sections may be optimal [40]. For cells of smaller size and/or volume, such as microglia, a 10–15 µm thickness is likely sufficient [39]. While cells isolated from mouse tissue may allow, in some cases, for thinner sections of 10–15 µm, our previous unpublished work indicated 10 µm thick sections had reduced quality and quantity of RNA within the same specimens compared with 20 µm thick sections when normalized for volume. When examining RNA-seq results for cell-specific gene expression, we found our LCM protocol for isolation of Nissl-stained layer III and layer V pyramidal neurons targets the expected excitatory neuron populations, with little signal from inhibitory or non-neuronal (including, but not limited to, astrocyte, oligodendrocyte, and microglial) gene expression. Crystal formation within the cells and on the tissue due to high humidity and warming of the tissues can cause cell degradation, leading to RNase release from intracellular compartments and increased RNase activity [41]. Nissl staining is a simpler staining protocol, with the benefit of less attrition of RNA quality [31], and, thus, can be used when morphology and spatial localization is sufficient for neuronal identification. Endogenous fluorescence can be utilized in mouse models expressing green fluorescent protein (GFP), mCherry, or other tagged proteins, although the loss of these soluble proteins due to diffusion from broken cells in unfixed tissue can reduce signal intensity of fluorescent proteins, reducing the ability to isolate individual neurons [42].

There are several different kits enabling reproducible RNA isolation from single cells or low cell numbers. Differences in commercially available kits include types of lysis buffer, the performance of a phenol–chloroform separation, columns and column substrates, as well as wash buffers and DNase digestion steps. Utilizing the miRNeasy kit allows for the isolation of total RNA, including miRNA, ncRNA, rRNA, tRNA, and mRNA. Sample quality can have profound effects on downstream applications and must be taken into consideration. In performing LCM, staining protocols, time to identify and collect samples, as well as storage conditions can all affect RNA integrity. RIN from 1–8 following LCM (RIN 1 = completely degraded, RIN = 10 completely intact) or DV_200_ [32] from 20–70% are standard depending on the aforementioned measures, meaning, in some cases, a significant percentage of the RNA collected is not fully intact; see Table 2: RNA Isolation and Library Preparation Tips and Tricks for suggestions on overcoming low-input and low-quality RNA. In addition to the Takara Library preparation kit, Illumina also offers kits for formalin-fixed paraffin-embedded (FFPE) tissues or other tissue sources where RNA is significantly degraded due to fixation and/or tissue processing. The TruSeq RNA Exome has a limit of 10 ng of starting material, whereas the Illumina Stranded Total RNA prep with Ribo-Zero Plus is rated for as little as 1 ng of starting RNA. Several other kits allow for the introduction of RNA-seq library preps from degraded samples. However, the limitation, regardless of the isolation kit, is the actual starting material [43].

## 5. Conclusions

In conclusion, we detail a protocol to successfully microisolate individual adult neurons from *in vivo* brain tissues that are suitable for spatial transcriptomic analysis. Several key steps within this pathway can be modulated during sample preparation to utilize this protocol for samples from colorimetric, antibody-stained tissue, or fluorescently tagged cells. We document pitfalls, caveats, and salient points to consider to successfully navigate from brain collection to RNA-seq library preparation in the context of single-population assessments. Importantly, this protocol can be utilized for mouse or human neuron purification and has been successfully employed for antibody, colorimetric, and fluorescently tagged neuron isolations by LCM. Other species have not been investigated, but the likelihood of success is high based on our assessments of optimally (e.g., mouse) and suboptimally (e.g., human) prepared and collected brain tissue samples. Different cell types outside the CNS (e.g., other peripheral cell types and cancer cells) are also likely to be excellent input sources of RNA, considering the difficulty associated with neuronal isolation and the success we have demonstrated herein. Finally, we highlight some ‘tips and tricks’ that may help to troubleshoot problematic dissections or isolations.

## Figures and Tables

**Figure 1 jcm-12-03304-f001:**
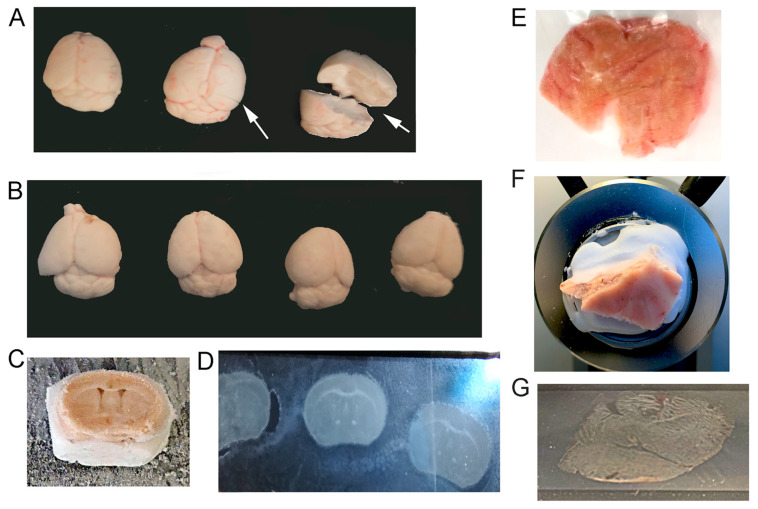
Preparation of brain specimens for LCM. (**A**) Mouse brains frozen using liquid nitrogen are represented with cracks or breaks (see arrows). (**B**) Powdered dry ice or an ethanol/dry-ice slurry preserves brains quickly without distorting brain tissue. (**C**) Frozen, unfixed mouse brains are prepared by mounting with OCT on the cryostat quick freeze shelf at −20 °C prior to sectioning. (**D**) Cryostat sectioning is performed at 20 µm and adhered to PEN or PET membrane slides. (**E**) Postmortem human brain tissue (frontal cortex; Brodmann area 9) is trimmed to fit PEN membrane slides. (**F**) Postmortem human brain tissue is mounted on disk with OCT with additional OCT around areas that are loose or cracked. (**G**) Single tissue sections are mounted on PEN membrane slides at 20 µm thickness for pyramidal neuron collection.

**Figure 2 jcm-12-03304-f002:**
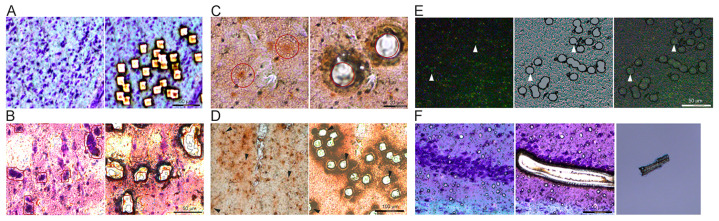
Representative images illustrating LCM of individual neurons. (**A**) Mouse Nissl-stained neurons are collected individually and isolated using the ‘point-to-point’ feature. (**B**) The ‘automated detection system’ was used to identify individual or small clusters of neurons in postmortem human frontal cortex tissue. Black line indicates automated detection limit, while red line indicates cut line (~15 pixels or ~6 µm out from detection limit using 20× objective) to ensure no cutting through edges of the neurons selected. (**C**) Two identified and marked ChAT-immunoreactive neurons at 40× objective before cutting, with blank areas showing where laser-cut cell and cell dropped into lysis buffer. (**D**) ChAT-immunoreactive neurons are shown before and after microisolation, with neurons indicated by arrows not completely cut and dropped into collection device, which were removed from cell counts. A halo of burned edges indicates offset was slightly out of focus and needed to be adjusted for this area of the tissue, which can be common when cutting cells over a large region of tissue. (**E**) Mouse fluorescently tagged neurons on PET slides are identified under fluorescence (**left**) but cut under brightfield (**middle**). Overlay (**right**) of these two images is critical to determining cutting of neurons correctly. White arrows show cutting of select neurons in each field of view. (**F**) Tightly packed mouse hippocampal CA1 pyramidal neurons were cut in ribbon swaths of cells, reducing the edges cut by the laser and time needed for collection, both of which have a beneficial effect on RNA quality. The PEN membrane of these larger sections readily seen in collection device (right) even though cells are digested by lysis buffer. Scale bars: 50 µm (**A**,**B**,**E**); 100 µm (**D**,**F**); 20 µm (**C**).

**Figure 3 jcm-12-03304-f003:**
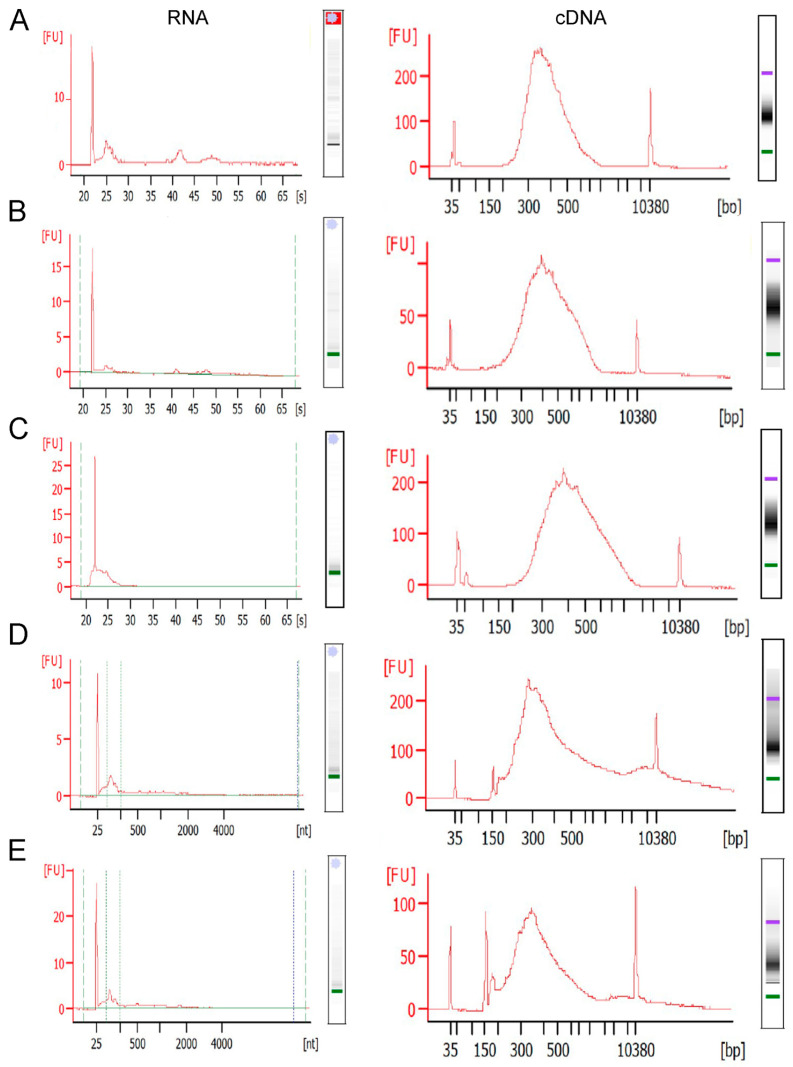
Representative bioanalysis traces for RNA extraction and cDNA library preparation from neurons acquired via LCM depending on cell input and cutting parameters. (**A**) ~550 Nissl-stained neurons from the mouse CA1 hippocampal region were cut in a ribbon and purified for RNA-seq (**left**); RNA purification gave RIN of 4.5, with cDNA library from RNA performed with 4 min fragmentation, 10 µL starting RNA (equivalent of ~375 neurons; (**right**)). (**B**) ~1000 Nissl-stained neurons from the mouse CA1 hippocampal region were isolated by LCM in small clusters; RNA resulted in RIN of 1 (**left**), with cDNA library performed with 2 min fragmentation (low RIN), 8 µL starting RNA (~570 neurons; (**right**)). (**C**) ~500 ChAT-immunoreactive neurons from the mouse basal forebrain were isolated individually, with RNA purification giving a RIN 1.5; cDNA library was performed without fragmentation (option 2) with entire RNA volume (excepting 1 µL for quantification) of starting RNA used (**right**). (**D**) ~500 Nissl-stained postmortem human neurons from BA9 (layer V) were isolated individually using the automated detection system for LCM, with RNA purification yielding a DV_200_ of 47.5 (**left**); half of RNA sample was used (~250 neurons) for cDNA library, which was performed with 2 min fragmentation (**right**) (**E**) ~775 Nissl-stained human neurons from BA9 layer V were isolated individually using the automated detection system for LCM, with RNA purification yielding a DV_200_ of 50 (**left**); 8 µL of RNA sample was used (~475 neurons), and cDNA library was performed with 2 min fragmentation (**right**).

**Table 1 jcm-12-03304-t001:** Laser capture microdissection: tips and tricks.

Laser Capture Microdissection: Tips and Tricks
Prior to experimental onset, determine the best fit for slide membrane thickness and composition for the planned experiment.
Polyethylene naphthalate (PEN) 4 µm slides are the most common for use with the LMD7000. Polyethylene terephthalate (PET) 0.9 µm and polyphylene sulfide (PPS) 4.0 µm steel frame slides have lower (than PEN) autofluorescence, with the PET working best for our applications upon rigorous testing. Ropolymer (Fluo) steel frame slides are considered the best for little to no autofluorescence but are expensive. Different fixation protocols may hinder RNA quality to varying degrees. Common fixatives of acetone, ethanol, DSP, paraformaldehyde, or formalin add an experimental variable.
Differences in membrane thickness, as well as glass versus steel frame slides, will affect placement and laser cutting.
Be sure to place slide correctly on the stage to ensure cut tissue falls properly into the collection device. Test laser cutting settings for each slide or section to correct for minute differences in section or slide thickness, which may cause ablation or burning of tissue. The ‘laser screw’ feature allows to cut thicker sections without increased power or aperture of the laser. Automated detection allows for cutting of irregularly shaped cells and is a rapid method of identification when background is low and may cut down on experimenter error.
Objectives
Lower power objectives (5 and 10×) may be used for orientation, overview, or for regional dissections. Higher power (20 or 40×) is optimal for single-cell dissections, with 63× or 100× used for small cells or subcellular dissections. Decreased objective power results in increased laser aperture, higher power settings, and possible inclusion of extraneous tissue.

**Table 2 jcm-12-03304-t002:** RNA purification and library preparation: tips and tricks.

RNA Purification and Library Preparation: Tips and Tricks
Keeping samples cold and avoiding freeze/thaw cycles will preserve RNA integrity. Utilize the DNase digestion step, as using random hexamers can result in genomic DNA amplification during library preparation. For brain tissue, kits that eliminate lipids with a phenol–chloroform step can help prevent clogging or poor elution. The number, type of cell, and collection of multiple cells in tightly packed structures will have an effect on total RNA isolated. RNA analysis by a bioanalyzer, rather than a spectrophotometer, will give an accurate assessment of both the quantity and quality of RNA isolated: ○ Agilent 2100 has an accuracy limit of ~50 pg/µL, but RIN values are unreliable below ~100 pg/µL. ○ Agilent Tapestation has an accuracy limit of ~100 pg/µL but will only give accurate RIN values at ~500 pg/µL. ○ In cases where RIN is unavailable, DV_200_ values are useful to estimate RNA quality. Library preparation from LCM with a polyT primer for reverse transcription eliminates degraded RNA from the amplification procedure. This approach has been unsuccessful in using antibody-stained mouse tissue in our laboratory.

## Data Availability

Data is available upon request from the authors.

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
