# Peer review of "Microisolation of Spatially Characterized Single Populations of Neurons for RNA Sequencing from Mouse and Postmortem Human Brain Tissues"

_jcm, 2023, doi:10.3390/jcm12093304_

Round 1

Reviewer 1 Report

In the manuscript, the researchers described a new method based on LCM, to isolate intact single neurons for RNA-seq. Overall the draft is in good shape. Some minor revisions may improve the quality of paper.

Minor concerns:

1. Author affiliation #1 needs editing.

2. The number that comes after each keyword should be deleted. 

3. The authors should briefly introduce the RNA quantification and purity examination method as described in citation 32. 

Author Response

Thank you very much for the thoughtful and thorough review of our manuscript JCM-2288979 “Microisolation of spatially characterized single populations of neurons for RNA-sequencing from mouse and postmortem human brain tissues” by Melissa J. Alldred and Stephen D. Ginsberg for consideration for publication as an Article in the Journal of Clinical Medicine. We are pleased to respond in a point-by-point fashion below.

Reviewer #1. We thank the Reviewer for her/his insightful comments and have emended the manuscript accordingly.

Minor comments:

1. "Author affiliation #1 needs editing."

Response: Corrected.

2. "The number that comes after each keyword should be deleted." 

Response: We removed numbers from each keyword.

3. "The authors should briefly introduce the RNA quantification and purity examination method as described in citation 32." 

Response: We have added a sentence (lines 242-243) to clarify the DV200 method.

Reviewer 2 Report

Alldred and Ginsberg have written a practical guide to isolating single cells (neurons) from brain tissue.  This guide addresses and advises many of the considerations one would need to address in working through a study utilizing this technique.  Nothing replaces hands-on experience, but this guide facilitates entry to this method and brings focus to points that are not immediately obvious as warranting careful attention.

The main considerations come down to sectioning tissue, cryogenic considerations, cell labeling, laser capture microdissection, RNA isolation and RNA library preparations.  The most detailed emphasis is placed on the former items, which is reasonable considering the most upstream steps ultimately have the greatest influence on quality of the downstream results.  The detailed and clear methodology with respect to RNase free and cold temperature techniques is necessarily detailed.  Importantly, the earlier steps are most broadly conserved across any study looking to capture neurons, while the latter are somewhat instrument and choice-of-kit specific.

This reviewer finds little to critique with regard to the core content of the manuscript.  Comments or critiques that are most pertinent follow, with some minor specific corrections below.

- At least in this reviewer’s typeset version, the images in Figure 2 should be larger in order to see the required detail.  

- While there is no set thickness to cut tissue at, the only other point that I was not in full agreement with was guidance to section tissue at 20 um.  My inclination was that this was too thick - however, the justification and reasoning (lines 240-244) is thoughtful.  I would caution that this thickness increases risk of capturing overlapping cells.  While stated typical cell thicknesses are appropriate, cutting at this thickness assumes a section exactly even to the upper plane of the cell.  In reality, most cells will be cut within the cell itself, so the thicker sectioning is in my opinion excessive.  Although, the points at lines 340-350 may justify the trade-off of thicker sections for higher quality RNA.  

Other specific comments:

Line 13: This sentence in the abstract did not read well.  I am not sure if it was meant to read something like "Recent studies indicate unique transcriptomic profiles for spatially localized neurons."

Line 210 requires indentations whereas lines 225, 336, 338, and 343, 354 have indentation issues.

Line 282: The degrees Celsius symbol looks off compared to other instances.

Line 292-293 - Please confirm that the statement regarding Fig 3A-C have less RNA than 3D-E.  My interpretation of the left-side Bioanalyzer traces would be the opposite.  It is difficult to tell by eye and the software is probably integrating the smallest fragments next to the left marker, so it may be true.  It is difficult to tell at very low input if the lack of rRNA integrity is due to minimal input or overall degradation. If the statement is based off the software concentration output, then it is fine, but still difficult to connect to the figure panels themselves.  Also, have the authors considered commenting on the probable adapter dimers in 3E (and 3D)?

Line 321 - Should it be "...were purified for RNA-seq (left) and cut in bulk, RNA ..."?

Line 404: The statement is missing a word - should it be "Ethical review and approval were waived for human tissue for this study..."?

Supplemental file.  This is useful as a printable lab method.  Step viii of rapid antibody staining protocol has an extra space in "1 ml pipette".

Author Response

Reviewer #2. We thank the Reviewer for her/his helpful commentary and have emended the manuscript in response to these inquiries.

Major Comments:

1. “At least in this reviewer’s typeset version, the images in Figure 2 should be larger in order to see the required detail”.

Response: A high resolution (600 dpi) copy of Figure 2 has been uploaded. Due to sizing constraints of the journal the figure inset is by default smaller. Actual size, high resolution images are also available by request to the authors

2. “I would caution that this thickness increases risk of capturing overlapping cells… Although, the points at lines 340-350 may justify the trade-off of thicker sections for higher quality RNA.”

Response: This is an important point. Our previous and current single population RNA-seq results demonstrate that we do not capture RNA from overlapping non-neuronal cells at this section thickness. We added an additional sentence in the Discussion (lines 423-427) to highlight our findings.

Minor Points:

1. “Line 13: This sentence in the abstract did not read well.” 

Response: We revised this sentence for clarity.

2. “Line 210 requires indentations whereas lines 225, 336, 338, and 343, 354 have indentation issues.”

Response: Corrected as requested.

3. “Line 282: The degrees Celsius symbol looks off compared to other instances.”

Response: Corrected as requested.

4. “Line 292-293 - Please confirm that the statement regarding Fig 3A-C have less RNA than 3D-E. … Also, have the authors considered commenting on the probable adapter dimers in 3E (and 3D)?”

Response: We added text confirming the RNA concentration (lines 356-359). For the library traces (Fig., 3D & E), we provided additional discourse on the detection of adapter dimers (lines 254-256).

5. “Line 321 - Should it be "...were purified for RNA-seq (left) and cut in bulk, RNA ..."?”

Response: This sentence was revised.

6. “Line 404: The statement is missing a word”

Response: Fixed. Thank you for catching this error.

7. “Supplemental file. This is useful as a printable lab method. Step viii of rapid antibody staining protocol has an extra space in "1 ml pipette".”

Response: Corrected as requested.

We thank the Reviewers and Editorial staff in advance for their time and effort dedicated to reviewing our submission which we feel is worthy of publication in the Journal of Clinical Medicine.

Round 2

Reviewer 1 Report

Thanks for the revision. The manuscript is in good shape now.

Reviewer 2 Report

The edits made in response to the initial review address any questions or critiques that were raised by this reviewer.